

# Comparing multiple comparisons: practical guidance for choosing the best multiple comparisons test

Stephen Midway[1], Matthew Robertson[2], Shane Flinn[3] and Michael Kaller[4]

[1] Department of Oceanography and Coastal Sciences, Louisiana State University, Baton Rouge, LA, United States of America
[2] Centre for Fisheries Ecosystems Research, Fisheries and Marine Institute of Memorial University of Newfoundland, St. John's, Newfoundland and Labrador, Canada
[3] Quantitative Fisheries Center, Department of Fisheries and Wildlife, Michigan State University, East Lansing, MI, United States of America
[4] School of Renewable Natural Resources and Department of Experimental Statistics, Louisiana State University Agricultural Center, Baton Rouge, LA, United States of America

Corresponding author
Stephen Midway, smidway@lsu.edu

## ABSTRACT

Multiple comparisons tests (MCTs) include the statistical tests used to compare groups (treatments) often following a significant effect reported in one of many types of linear models. Due to a variety of data and statistical considerations, several dozen MCTs have been developed over the decades, with tests ranging from very similar to each other to very different from each other. Many scientific disciplines use MCTs, including >40,000 reports of their use in ecological journals in the last 60 years. Despite the ubiquity and utility of MCTs, several issues remain in terms of their correct use and reporting. In this study, we evaluated 17 different MCTs. We first reviewed the published literature for recommendations on their correct use. Second, we created a simulation that evaluated the performance of nine common MCTs. The tests examined in the simulation were those that often overlapped in usage, meaning the selection of the test based on fit to the data is not unique and that the simulations could inform the selection of one or more tests when a researcher has choices. Based on the literature review and recommendations: planned comparisons are overwhelmingly recommended over unplanned comparisons, for planned non-parametric comparisons the Mann-Whitney-Wilcoxon $U$ test is recommended, Scheffé's $S$ test is recommended for any linear combination of (unplanned) means, Tukey's HSD and the Bonferroni or the Dunn-Sidak tests are recommended for pairwise comparisons of groups, and that many other tests exist for particular types of data. All code and data used to generate this paper are available at: https://github.com/stevemidway/MultipleComparisons.

## INTRODUCTION

Many of the popular and robust statistical techniques used in data analyses estimate group (or treatment or factor level) means. Data analyses are crowded with factors of interest from experiments and observations in which different groups show different effects and responses—and these significant results are what progress scientific knowledge. Models for

evaluating the existence of differences among means include a wide range of linear models. The classic ANOVA (ANalysis Of Variance) is a general linear model that has been in use for over 100 years (*Fisher, 1918*) and is often used when categorical or factor data need to be analyzed. However, an ANOVA will only produce an $F$-statistic (and associated $p$-value) for the whole model. In other words, an ANOVA reports whether one or more significant differences among group levels exist, but it does not provide any information about specific group means compared to each other. Additionally, it is possible that group differences exist that ANOVA does not detect. For both of these reasons, a strong and defensible statistical method to compare groups is nearly a requirement for anyone analyzing data.

The lack of specifically being able to compare group means with ANOVA has long been known and a sub-field of *multiple comparisons tests* (MCTs) began to develop by the middle of the 20th century (*Harter, 1980*). Of course, when the analysis only includes two groups (as in a $t$-test), then a significant result from the model is consistent with a difference between groups. However useful this approach may be, it is obviously very limiting and as *Zar (2010)* states: "employing a series of two-sample tests to address a multisample hypothesis, is invalid." What has developed over the last several decades has been a bounty of statistical procedures that can be applied to the evaluation of multiple comparisons. On the surface, this long list of options for MCTs is a good thing for researchers and data analysts; however, all the tests are unique, and some are better suited to different data sets or circumstances. Put another way, some MCTs are questionable or invalid when applied to certain experimental designs and data sets. Additionally, because MCTs are applicable to general linear models, generalized linear models, hierarchical models, and other models, both the data and the model need to be considered when selecting an MCT.

Because a large proportion of scientists that use MCTs are not statisticians or otherwise versed in the details and nuance that inform their application, there are numerous cases where MCTs are used incorrectly. For example, *Ruxton & Beauchamp (2008)* reviewed 12 issues of *Behavioral Ecology* and reported on 70 papers that employed some type of multiple comparisons testing (or homogeneity of means, as they report). Their review found 10 different types of MCT used, including "by eye" and "undefined." It would not be surprising to learn that similar inconsistencies in the application of MCTs exist in other fields, and it is clear that use and reporting of MCTs in many scientific disciplines is far from standardized. Although non-statisticians need to use some criteria for selecting and reporting an MCT, some confusion may be expected as the statistical literature does not always provide non-technical recommendations for strengths, weaknesses, and applications of MCTs.

Two factors motivated our study. First, scientists across disciplines are unlikely to use technical statistical literature to self-train on the correct use and best practices for MCTs. And second, scientists across disciplines are certain to continue using MCTs. Given these two conditions, we identified a need for non-statisticians to be better equipped to make decisions about which MCTs to use under different circumstances. Even when a correct MCT is used, a better understanding and clearer reporting of the application of the test may be warranted and may also improve the reporting.

**Table 1  Four outcomes of hypothesis testing.** The two types of error are presented in boldface text.

|  | Null Hypothesis ($H_0$) True | Null Hypothesis ($H_0$) False |
| --- | --- | --- |
| Fail to reject | Correct (true negative) | **type II error (false negative)** |
| Reject | **type I error (false positive)** | Correct (true positive) |

The objectives of this study are to: (1) Identify the most common MCTs used historically and currently in published ecological literature, (2) Conduct a simulation whereby the most commonly used tests are evaluated based on data sets with different (but common) attributes, (3) Evaluate data-independent considerations (e.g., planned vs. unplanned tests) for MCT selection, and (4) Combine the results of the simulation study with known best-practices for MCTs to arrive at recommendations for their selection and use.

## Background on the conditions of multiple comparison

The development of MCTs has been undertaken to address the lack of specificity in comparing group means when using other statistical tools (e.g., ANOVA); however, all MCTs attempt to address the same inherent problem that stems from the propagation of statistical errors in hypothesis testing. Recall that the basic design of hypothesis testing yields one of four possible outcomes, which are the product of two possible states of the null hypothesis and two possible decisions about the null hypothesis (Table 1). For the most part, we can ignore the two correct inferences that occur when the null hypothesis is true and we fail to reject it, or when the null hypothesis is false, and we correctly reject it. These are desired outcomes. The outcomes we need to be concerned with are commonly referred to as errors: a type I error is a false positive, or rejecting a true null hypothesis, and a type II error is a false negative, or failing to reject a false null hypothesis. Type I error is also commonly notated with $\alpha$ and type II error with $\beta$, both of which can be thought of as probabilities.

Any test statistic and associated *p*-value is inherently providing an inference on an outcome. Because statistics involves describing quantities from distributions and their associated variability and uncertainty, statistical tests are not confirmatory by nature, and rather attempt to provide a level of confidence about the outcome or estimate. Herein lies the challenge with MCTs. Any singular statistical outcome is compared against an $\alpha$, or *a priori* significance threshold. $\alpha$ is often selected based on convention (e.g., 0.05), but also represents a compromise between a rigorous amount of evidence required (to avoid type I error), but not so much evidence that we would never find a statistically significant result (to avoid type II error). We know that $\alpha$ will occasionally let a false positive go, but when conducting singular statistical tests, we accept this risk and have it quantified. With multiple comparisons, we are artificially increasing the number of tests (or test statistics), which greatly increases the chances of false positives if $\alpha$ is not adjusted. For example, if we run just 15 MCTs (which represents all the pairwise combinations of six groups), and we do not adjust $\alpha$, our probability of a type I error is over 50%. The solution is to reduce $\alpha$ such that there is a higher significance threshold that should reduce the false positives. However, when $\alpha$ decreases, our $\beta$ will increase as we are likely to misdiagnose null hypotheses that are false. At some level, all MCTs are an attempt to balance $\alpha$ and $\beta$

based on various criteria, including aspects of the data and the number of comparisons to be conducted.

The error-rate adjustments used in MCTs have their own terminology, which we will use in this study. Specifically, pairwise comparison error rate (PCER) is the probability of committing an error for an individual comparison and is often the error rate referred to in orthogonal comparisons (see below), or in cases where error-rate adjustments do not have to be made. The experimentwise type-I error rate (EER) is the error rate that reflects the probability of at least one type-I error occurring in a situation where several independent comparisons are made (the example in the above paragraph). The EER reflects the adjustment in PCER to account for multiple comparisons, and the variability in ways to adjust the PCER accounts for the variety of MCTs available. Additionally, MCTs that control the EER to below 5% (by strongly reducing PCERs) are known as *conservative*, while those with less strong adjustments of the PCERs which do not control the EER to at or below 5% are known as *liberal*. Finally, the term familywise error rate (FWER) is also commonly used to describe EER, and while this study uses them interchangeably, FWER and EER in some cases may be used to describe different collections of comparisons.

## MATERIALS & METHODS

### Literature review of multiple comparisons test usage

A literature search was conducted on 14 August 2019 to count the numbers of putative uses of 17 different MCTs reported in the ecological literature (Table 2). The search was conducted in Google Scholar because Web of Science (and comparable literature search programs) do not search the full text of articles, and MCTs are not commonly included in the title, abstract, or other searchable article information; MCTs are typically reported in the main body of the text. Although Google Scholar searches the whole text, there are some search limitations. For example, we were not able to search by discipline or category of journals. As a workaround, we specified in our search parameters that only journals with the term *ecology* in the title be searched. We recognize that this is an imperfect method to exhaustively search the full discipline of ecological journals; however, it was a bias imposed on all searches and the results likely included enough journals that we expect to have found the general trends. MCTs are also used in many fields beyond ecology; however, we wanted a field that was large enough to likely have all tests represented, while still confining our search to a specific scientific field. (And the authors all operate within the domain of ecology, so we felt most comfortable working with this literature.)

As stated earlier, we also recognize that some MCTs have different names or abbreviations. For Tukey's Honest Significant Difference (HSD) and Fisher's Least Significant Difference (LSD) we searched by abbreviation, under the assumption that while any first mention of a test would include the whole test name, these specific abbreviations are well-established and very common. Therefore, we examined different terms for test names (e.g., Tukey's HSD vs. Tukey's Test vs. Tukey) and ultimately searched the term that we thought was the most descriptive of the test (although in instances where we searched multiple terms, the search results were typically similar). We could not exclude

**Table 2** **Multiple comparisons tests (MCTs) searched in the literature and total number of reported uses from 1960–2019.** Tests are ordered by their general application and then by popularity—the number of times cited in ecological literature. Note: Terms like *test* and *procedure* have been removed where not necessary. Based on the literature, the bottom three tests are often not recommended, which is guidance we have adopted (and discuss in the study). Finally, we are not able to differentiate Bonferroni from sequential Bonferroni, but we expect that the number of reported citations captures most of both uses.

| Test | Citations ($n$) | General application |
| --- | --- | --- |
| Bonferroni | 20,801 | Parametric situations |
| Tukey's Honest Significant Difference (HSD) | 7,800 | Parametric situations |
| Tukey-Kramer | 1,930 | Parametric situations |
| Scheffé's $S$ | 1,370 | Parametric situations |
| Dunn-Šidák (Šidák) | 905 | Parametric situations |
| Dunnett's | 332 | Parametric situations |
| Ryan's | 108 | Parametric situations |
| Waller-Duncan $k$ | 103 | Parametric situations |
| Dunn Procedure | 2,870 | Nonparametric situations |
| Mann-Whitney-Wilcoxon $U$ | 1,950 | Nonparametric situations |
| Games-Howell | 248 | Nonparametric situations |
| Nemenyi | 184 | Nonparametric situations |
| Steel-Dwass | 94 | Nonparametric situations |
| Fligner-Policello | 8 | Nonparametric situations |
| Student-Newman-Keuls (SNK) | 1,800 | Not recommended |
| Fisher's Least Significant Difference (LSD) | 971 | Not recommended |
| Duncan's Multiple Range Test (DMRT) | 1,707 | Not recommended |

any instances where a specific name eponymous with an MCT (e.g., Šidák) could have been the name of an author or some other use of that name unrelated to multiple comparisons. Despite this limitation, most of the tests we searched included terms in addition to just a name and we expect any mentions of our search terms to be almost entirely related to multiple comparisons. In other words, it is unlikely that a person named after or sharing a name with an MCT would have published so much in the ecological literature that they would overwhelm the search of a common statistical test. We were interested in looking at the uses of certain MCTs over time, and as such, select tests terms were searched for in the literature in 5-year blocks, starting in 1960 and going to 2019.

## Simulation of multiple comparison tests

Although there are certain recommendations that describe how different MCTs should be used in different (experimental) design settings, there is not necessarily a correlation between experimental design (e.g., planned vs. unplanned comparisons) and the attributes of the subsequent data set (e.g., large or small sample size). In many design cases there will be options for MCTs and understanding sensitivities about the tests may inform their selection. Furthermore, test performance may be the determining factor when several tests are otherwise acceptable in a given situation. Therefore, to develop simulation-based recommendations for MCTs we evaluated the proportion of type I error, type II error, and the distribution of *p*-values for nine common MCTs under a range of data scenarios.

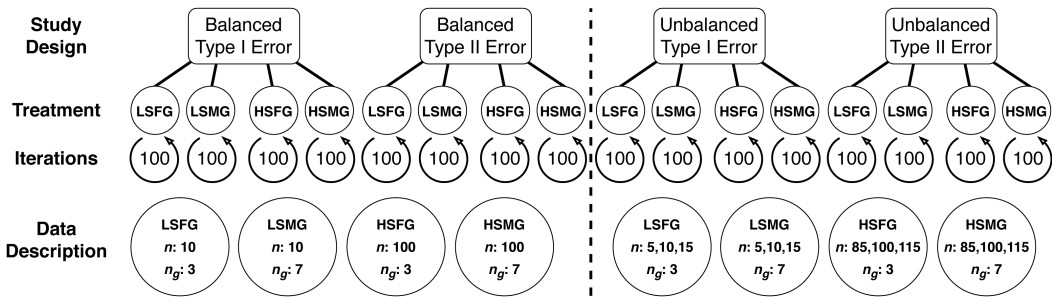

**Figure 1** **Diagram of multiple comparison simulations.** This diagram shows the organization of study design, treatment, iterations, and descriptions of the data used for each treatment in balanced and unbalanced designs (separated by the dashed line). The data description circles relate the treatment abbreviations to the number of samples in each group ($n$) and the number of groups ($n_g$). The abbreviations for the four simulation treatments are LSFG, low sample size with few groups; LSMG, low sample size with many groups; HSFG, high sample size with few groups; and HSMG, high sample size with many groups.

Overall, we evaluated four main types of simulation study designs: balanced study designs for (1) type I error and (2) type II error, as well as unbalanced study designs for (3) type I error, and (4) type II error. Balanced study designs involved simulations with the same number of samples for each group, while unbalanced study designs had groups with varying numbers of samples. Furthermore, type I error was assessed by simulating data with the same mean and standard deviation between all groups while type II error designs contained one mean that was different among all groups. Finally, within each study design we evaluated four simulation treatments: (1) low sample size with few groups (LSFG), (2) low sample size with many groups (LSMG), (3) high sample size with few groups (HSFG), and (4) high sample size with many groups (HSMG) (see Fig. 1).

## The data

A simulation iteration involved randomly drawing group samples from a normal distribution with a pre-specified number of samples, mean, and standard deviation. Given common practice for null hypothesis tests, we set group means to 0 except for type II study designs, where we set one group to have a mean of 1 rather than 0. Although in some datasets there may be multiple means that differ, our design was of a general and common scenario where a control group is different from treatment groups. We systematically tested simulations with increasing standard deviation values and determined that a standard deviation of 3 was an approximate threshold for providing contrast between tests. Furthermore, maintaining the same standard deviation among groups permitted consistency in the results. We chose group number and sample size values based on values that seemed appropriate given our experience with real (ecological) datasets. Low sample size simulations had 10 samples and high sample size simulations had 100 samples. Unbalanced group study designs involved low sample size groups with 5, 10, or 15 samples and high sample size groups with 85, 100, or 115 samples. Finally, simulations with few groups had 3 groups, while simulations with many groups had seven groups (see Fig. 1).

### The simulations

Simulations involved 100 iterations, where all MCTs were run for each iteration. Nine parametric MCTs were used to test for differences between groups, (1) Scheffé's *S* test, (2) *t*-test with Bonferroni correction, (3) *t*-test with Šidák correction, (4) Tukey's HSD, (5) Fisher's LSD, 6) Fisher's LSD with Bonferroni correction, (7) Fisher's LSD with Šidák correction, (8) Duncan's MRT, and (9) SNK. These nine tests were chosen based on their prevalence in the literature. We excluded the Dunnett's test because it is only applicable for special cases, and it would not have been appropriate to compare with other similar tests. Furthermore, we excluded Ryan's test due to the lack of readily available functions for its use, therefore limiting its applicability. We also excluded the Waller-Duncan *k* test because it does not use *p*-values to evaluate statistical significance and therefore would not be directly comparable to the other nine tests.

We systematically assessed results of our simulations with varied numbers of iterations. Results with more than 100 iterations did not significantly vary from results with 1000 iterations. We therefore chose 100 iterations for the sake of computational efficiency. When running type I tests (no group differences), we extracted estimates and their associated statistics for all groups and all simulations. When running type II tests (group differences), we extracted estimates and their associated statistics for group comparisons between groups with a mean of 1 and 0. We then evaluated the proportion of type I or type II errors and the distribution of *p*-values by MCT across all simulations. All simulations were run in *R Core Team (2020)* and examples of functions used for calculating the various MCTs can be found in Table 3. We have also included all the code needed to reproduce the results in this manuscript in Supplemental Code, found on the GitHub repository (https://github.com/stevemidway/MultipleComparisons) associated with this paper.

## RESULTS

### Literature review of multiple comparisons test usage

Our literature review reported 41,561 instances of 17 different MCTs from published studies in the field of ecology (Table 2). However, use of the different MCTs was very unbalanced. For instance, the Bonferroni (and sequential Bonferroni) procedure accounted for nearly half (20,801) of all MCTs used while six other tests were reported over 1,000 times. On the other extreme, the Fligner-Policello test was only reported eight times and was one of three tests that were not even reported 100 times. Our investigation into select tests over time reveals an overall increase in the use of MCT, but this use is almost entirely explained by a small number of very popular tests—namely Bonferroni and Tukey's HSD (Fig. 2).

One hypothesis about the observed frequency of reported MCTs is that if we assume tests are applied correctly, then their usage reflects the types of data and analyses that researchers are performing. For example, the extreme use of Bonferroni should reflect the commonness of parametric data and models, while the relatively less common nonparametric MCTs reflect fewer studies with nonparametric data and models. It might be expected that such large-scale interpretations are correct; however, the use and popularity of MCTs are likely influenced by several other factors, such as accessibility of tests in common statistical

**Table 3 Common multiple comparisons tests and their software implementations.** This table is meant to serve as a reference for functions and is not meant to advocate for particular packages and functions over others. Functions may give different results from one another, and we recommend reading any instructions or helpfiles for details on specific test implementations. Boldface functions indicate those used in the simulation component of this manuscript.

| Multiple comparisons test | R package::function | SAS Statements | SPSS Options |
|---|---|---|---|
| Tukey's HSD | **stats::TukeyHSD**[*] | MEANS / tukey; | Available by menu |
| | agricolae::HSD.test | LSMEANS / adjust = "tukey"[*]; | |
| | TukeyC::TukeyC | | |
| | DescTools::PostHocTest | | |
| $t$-test with Šidák correction | **MHTdiscrete::Sidak.p.adjust** | MEANS / sidak; | EMMEANS ADJ(SIDAK) |
| | mutoss::sidak | LSMEANS/ adjust = "sidak" | |
| $t$-test with Bonferroni correction | **stats::p.adjust** | MEANS / BON; | Available by menu |
| | mutoss::bonferroni | LSMEANS/ adjust = "Bon"; | EMMEANS ADJ(BONFERRONI) |
| Scheffé's $S$ | **agricolae::scheffe.test** | MEANS / scheffe; | Available by menu |
| | DescTools::ScheffeTest | LSMEANS / adjust = "scheffe"; | |
| | GAD::snk.test | | |
| | DescTools::PostHocTest | | |
| Student-Neumen-Keul's Test | **agricolae::SNK.test** | MEANS / snk; | Available by menu |
| | GAD::snk.test | LSMEANS /adjust = "snk"; | |
| | DescTools::PostHocTest | | |
| Fisher's LSD | **agricolae::LSD.test** | MEANS / LSD | Available by menu |
| | PMCMRplus::lsdTest | *not available in LSMEANS* | EMMEANS ADJ(LSD) |
| Fisher's LSD with Šidák correction | **agricolae::LSD.test** | *Not available* | *Not available* |
| | MHTdiscrete::Sidak.p.adjust | | |
| | mutoss::sidak | | |
| Duncan's MRT | **agricolae::duncan.test** | MEANS / duncan | Available by menu |
| | PMCMRplus::duncanTest | | |

**Notes.**
[*]as Tukey-Kramer.

software, familiarity or understanding of tests by non-statisticians (perhaps based on their simplicity or exposure from previous uses), and the overall power of the test. The issue of selecting an MCT based on power is not likely to be understood from the literature; however, it might be expected when more than one test is appropriate. For instance, a researcher may select the more liberal test with the expectation that increased power will produce (more) significant comparisons. This factor is likely at play to some degree, judging by the nearly 3,000 reported instances of SNK, Fisher's LSD, and Duncan's MRT, which are all tests that are not recommended based on inadequate error rate adjustment.

## Simulation

In general, the type I error for treatments with small and large sample size showed the reverse trends from one another when using balanced or unbalanced data (Fig. 3). For balanced data, LSFG treatments had more type I PCER than HSFG treatments (Fig. 3A). When there were many groups, type I PCER was greater with increased sample size for balanced designs, except for Duncan's MRT and Fisher's LSD. For unbalanced data, LSFG treatments had less type I PCER than HSFG treatments except for Scheffé's $S$ and LSMG

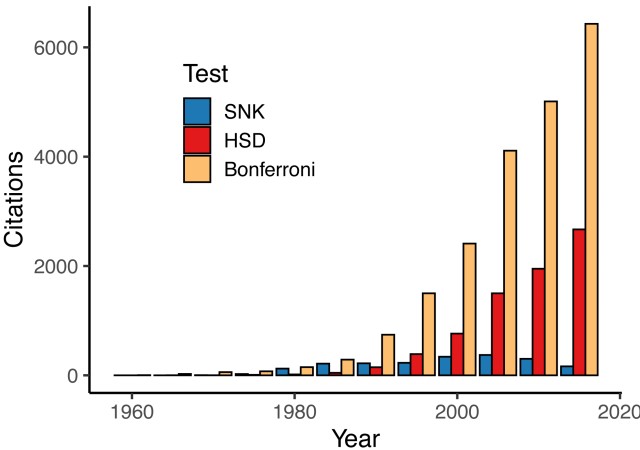

**Figure 2**  **Reported uses of the three most common parametric multiple comparisons tests (MCTs) by 5-year intervals.** Other MCTs excluded here (but listed in Table 1) show relatively similar trends to SNK or were too infrequently reported to visualize on this figure.

had more type I PCER than HSMG (Fig. 3B). When comparing between unbalanced and balanced study designs, all HSMG and LSFG tests had lower type I PCER, Duncan's MRT and Fisher's LSD had lower type I PCER for LSMG treatments, and the number of tests with lower type I PCER between balanced and unbalanced designs were split for HSFG treatments.

The Duncan's MRT and unadjusted Fisher's LSD tests provided the greatest proportion of type I error regardless of the study design or treatment (Fig. 3). The SNK test provided an equal or higher type I PCER than all tests other than Duncan's MRT and the unadjusted Fisher's LSD, although it never exceeded a PCER of 0.05. This was most noticeable in the density plots, where the SNK test did not have a peak in $p$-value density near one but instead was relatively constant from zero to one (Fig. 4). The Scheffé's $S$ test produced the least amount of type I error among all tests. The remaining five tests appeared almost identical in terms of proportion of type I error allowed (Fig. 3). All five tests had a large peak near one and a long tail to lower $p$-values (Fig. 4). Somewhat surprisingly, both tests using Bonferroni correction had the largest density near one for type I study designs, yet the long tail from these peaks resulted in Bonferroni having more type I error than Scheffe's $S$ test, which had a comparatively small peak near one. The density plots had similar trends between the balanced and unbalanced study designs, so only the balanced study design plots are shown (Fig. 4).

The patterns observed for proportion of type I error were approximately reversed for proportion of type II error due to the trade-off between these error rates (Fig. 5). Having high sample size was more important for type II error than having more groups. However, the distribution of $p$-values appeared to favor more type II error when there were more groups (Fig. 6). When sample sizes were large, the distribution of $p$-values was similar across all tests with peaks centered around or below 0.05 with tails of slightly varying sizes.

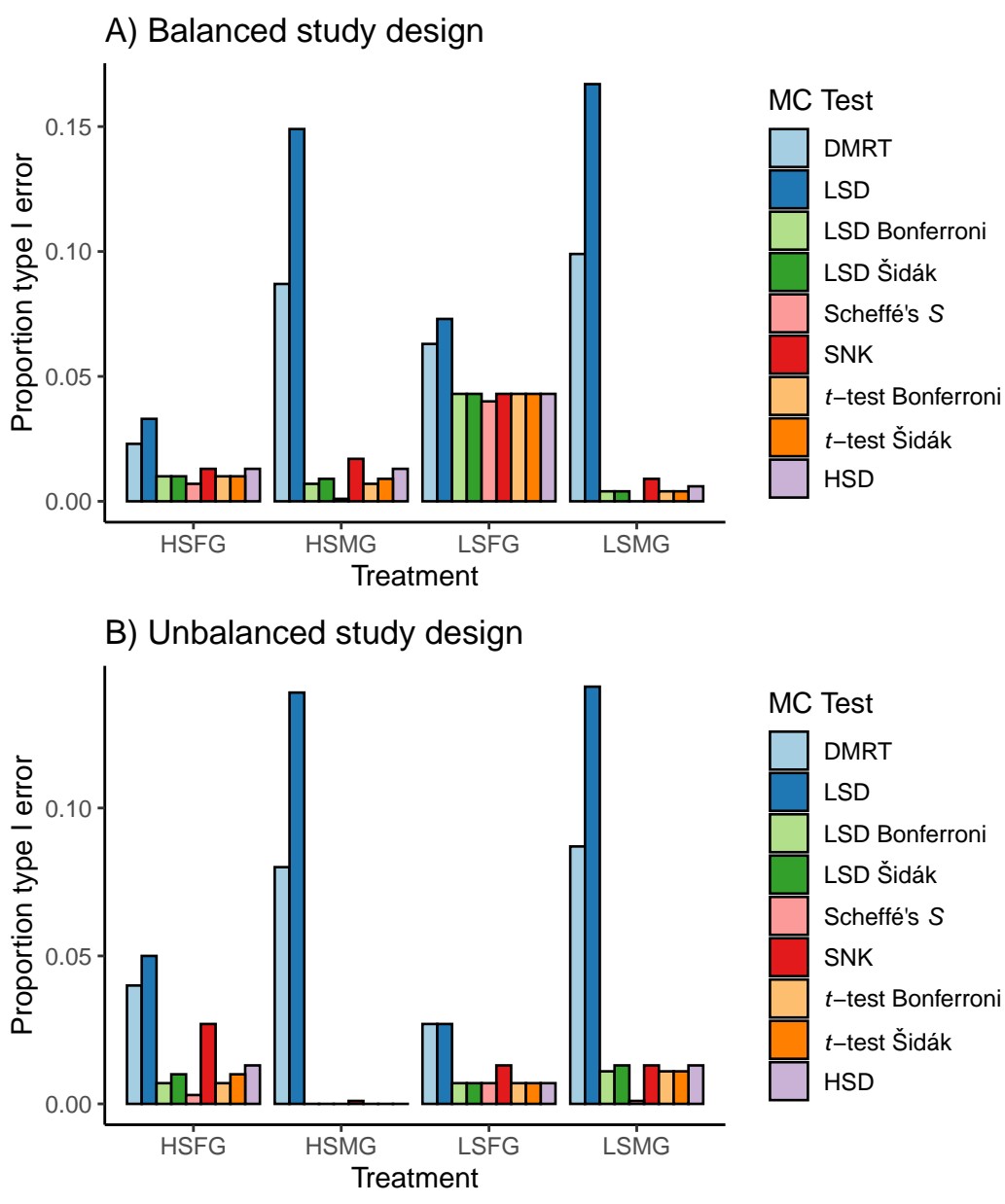

**Figure 3  Type I errors in simulations.** Proportion of type I per comparison error rates (PCERs) between the nine multiple comparison tests (MCTs) in each of the four simulation treatments for (A) balanced and (B) unbalanced study designs. Simulation treatment abbreviations can be found in the Fig. 1 caption.

Similar to the type I error tests, we have only shown the density plots for balanced study designs (Fig. 6).

## DISCUSSION

Our literature review and simulation were useful toward understanding how common MCTs performed under different—but realistic—data sets. The results from the simulation

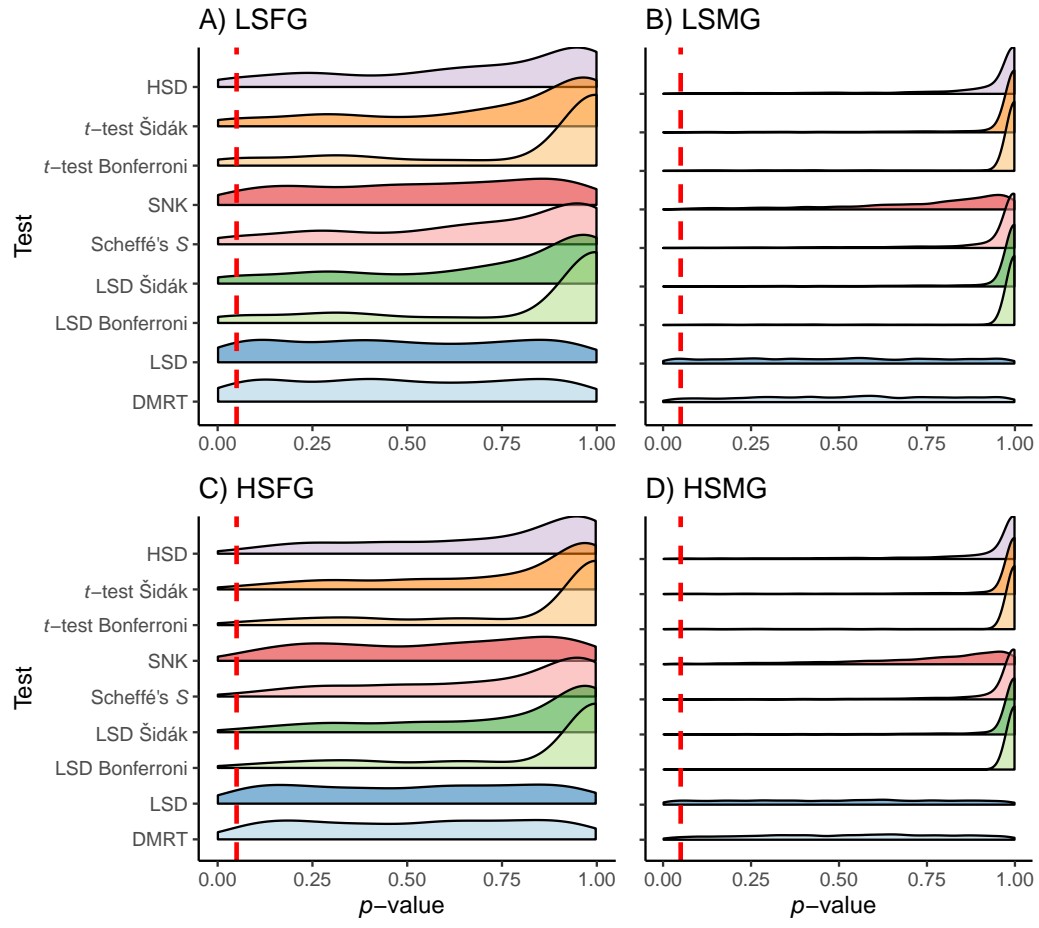

**Figure 4** *p*-values for type I errors in simulations. Distribution of *p*-values for type I error tests with balanced study designs (A–D). Distributions shown for the nine multiple comparison tests (MCTs) in each of the four simulation treatments. Dashed red line indicates a *p*-value of 0.05. Simulation treatment abbreviations can be found in the Fig. 1 caption.

may be useful toward helping ecologists decide which MCT(s) is right for their data and analysis; however, independent of the actual data there are other criteria that may inform the choice of test. Here, we review and summarize suggestions for the application of MCTs based on published studies. The structure of this section follows delineations of whether data are parametric or not and whether comparisons are planned or unplanned, both of which are dichotomies that are useful in selecting an MCT (see Fig. 7 for a diagram on selecting an MCT).

## Parametric or non-parametric data

The first delineation in MCTs is based on whether the data and model(s) are assumed to be parametric or non-parametric. As with several statistical decisions, knowing whether the data come from and exhibit parametric properties can greatly influence how the data need to be treated. Non-parametric data are immediately subjected to a different candidate

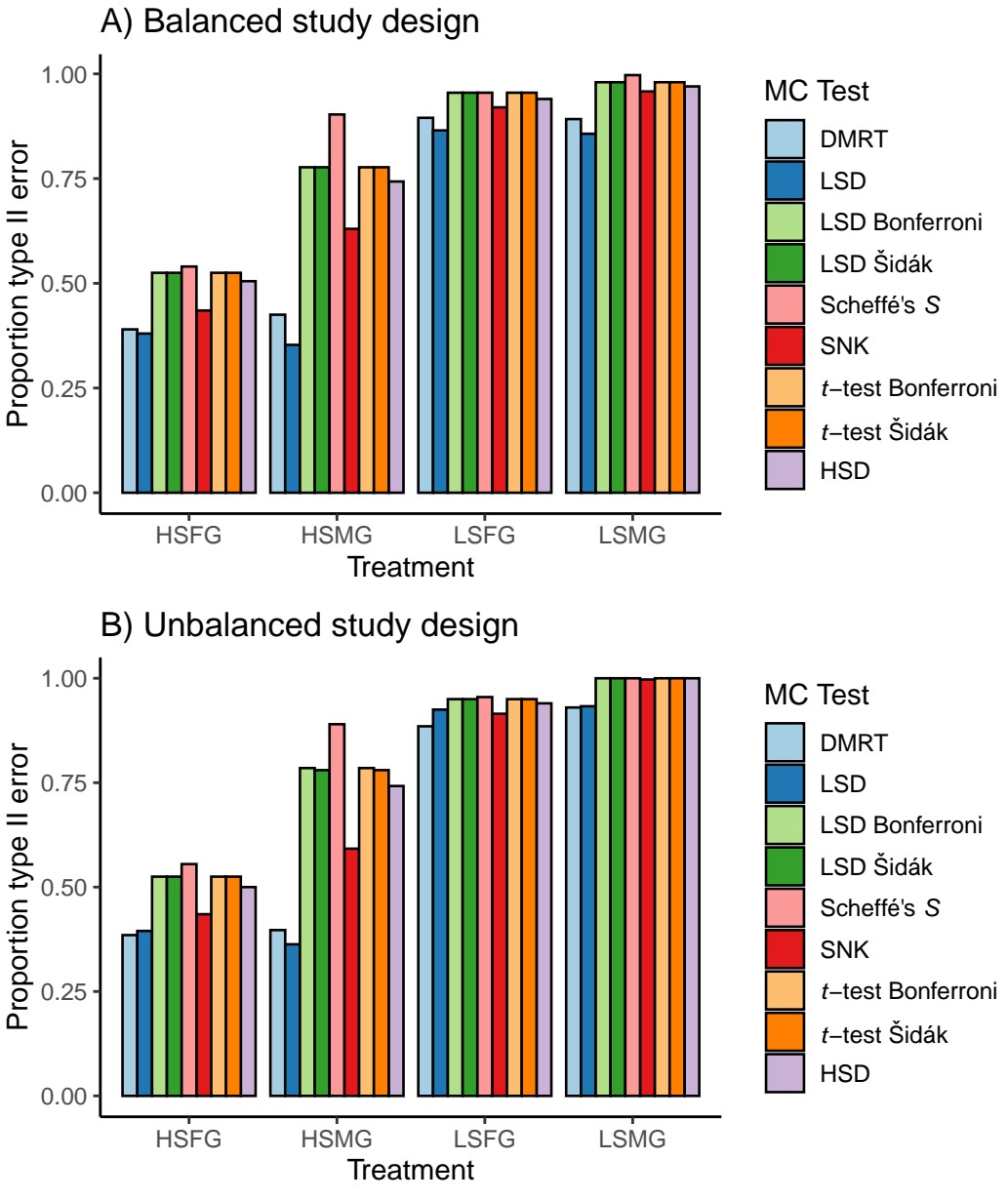

**Figure 5 Type II errors in simulations.** Proportion of type II per comparison error rates (PCERs) between the nine multiple comparison tests (MCTs) in each of the four simulation treatment for (A) balanced and (B) unbalanced study designs. Simulation treatment abbreviations can be found in the Fig. 1 caption.

list of MCTs than are parametric data (Fig. 7), although decisions still need to be made as there is not a universally recommended non-parametric MCT.

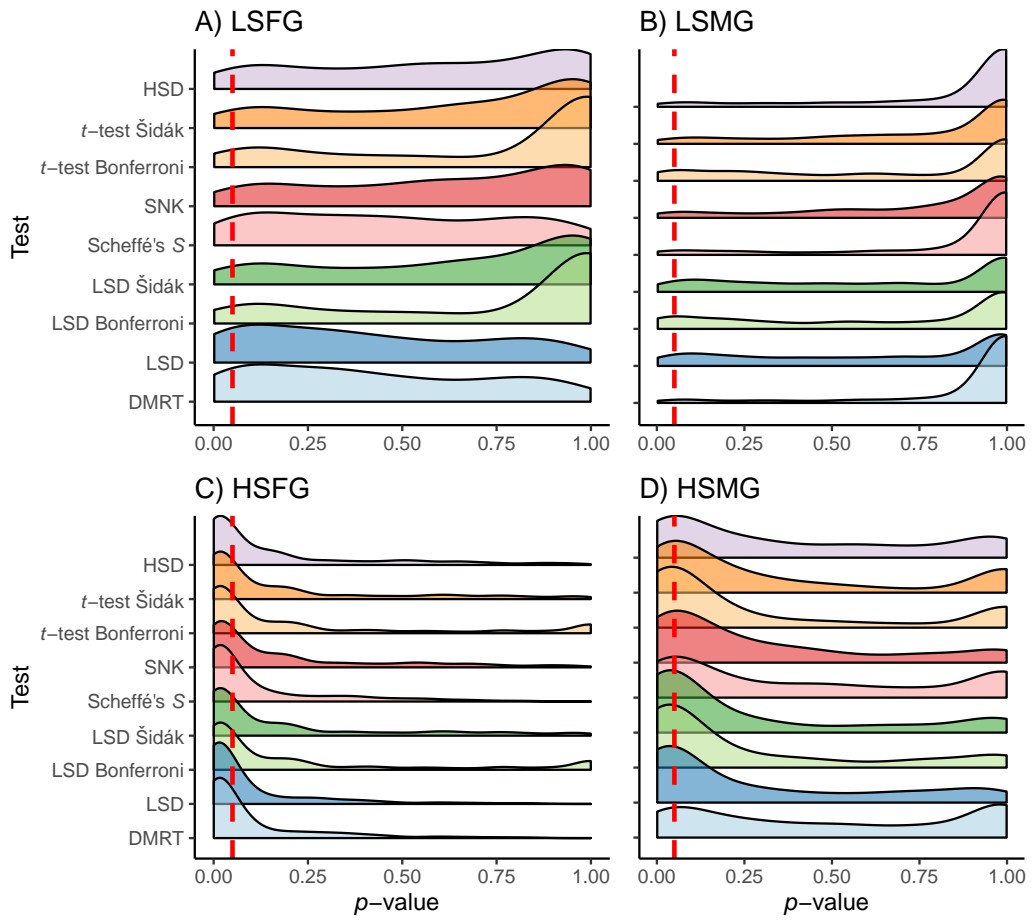

**Figure 6** *p*-values for type II errors in simulations. Distribution of *p*-values for type II error tests with balanced study designs (A–D). Distributions shown for the nine multiple comparison tests in each of the four simulation treatment for (A) balanced and (B) unbalanced study designs. Dashed red line indicates a *p*-value of 0.05. Simulation treatment abbreviations can be found in the Fig. 1 caption.

## Non-parametric MCTs

For non-parametric planned comparisons, the common Mann-Whitney-Wilcoxon *U* test is recommended (*Day & Quinn, 1989*), and if the distributions are symmetrical the Fligner-Policello test provides a robust alternative (*Fligner & Policello, 1981*). If non-parametric comparisons are unplanned, up to four different tests may be available. The Dunn procedure (*Dunn, 1964*) and Games-Howell Test (*Games & Howell, 1976*) are commonly used. If the order of rankings is an important consideration, then the Nemeyni Joint-Rank test (*Nemenyi, 1963*) is recommended for situations where all the data are ranked together (i.e., jointly), or alternatively, the Steel-Dwass test (*Steel, 1960*) is recommended for pairwise rankings, where data are re-ranked for each pairwise comparison.

It should also be mentioned that many researchers will use a non-parametric MCT if they think their residuals are not normally distributed. This could be a mistake because variances may still be unequal and because non-parametric tests are still valid for moderately

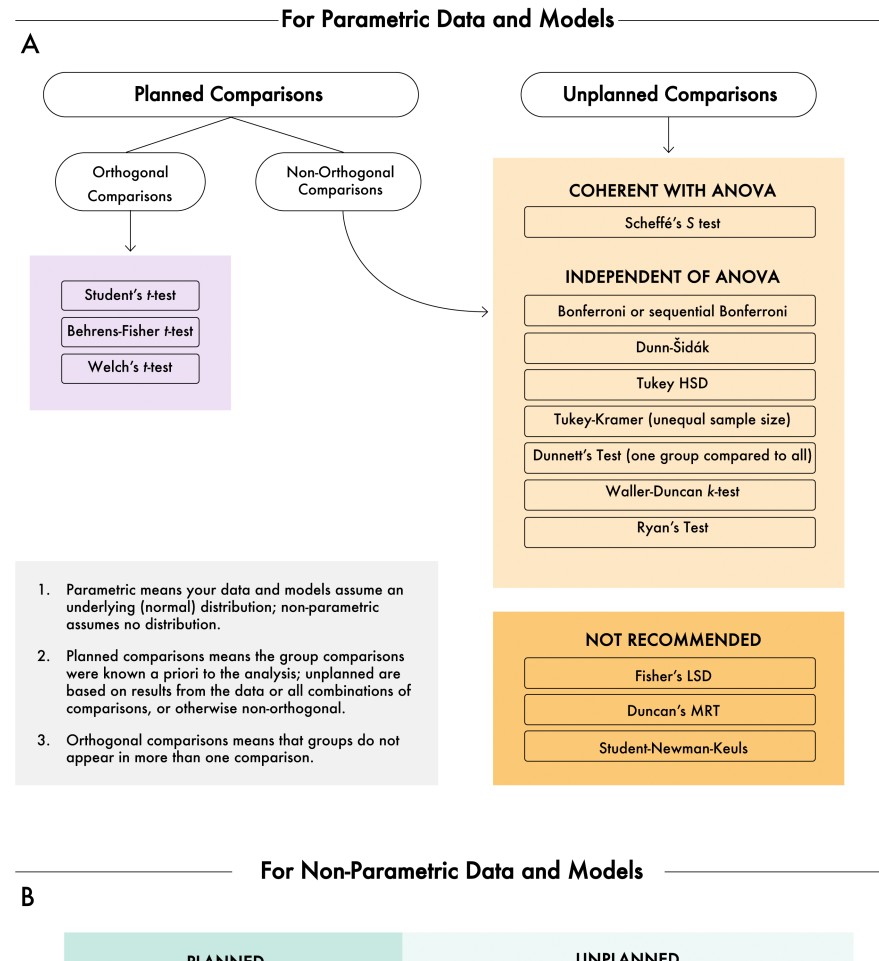

**Figure 7** **Decision diagram for selecting a multiple comparisons test.** (A) Multiple comparisons tests based on parametric data and models, followed by additional diagnostics. (B) Non-parametric data and models. The gray box defines some of the terms used in the diagram.

non-normal samples. However, a test like the Mann-Whitney-Wilcoxon $U$ test may be useful in a case of skewed data because the rank approach relaxes the effects of extreme values. For this reason, it is worth considering the features of a data set—and not just the outcome of a normality test—when considering MCTs.

## Parametric MCTs

If operating under a parametric assumption, planned comparisons may be the simplest option. Planned comparisons often represent the fewest number of comparisons, and if the comparisons are independent (i.e., orthogonal), allow a simple PCER (e.g., $\alpha = 0.05$) because there is no required adjustment to the EER. A Student's $t$-test can simply be

used when group variances are equal. A Behren's-Fisher $t$-test or Welch's $t$-test are recommended when group variances are unequal. The following sections focus primarily on MCT test options for unplanned parametric MCTs, although the conditions for planned and unplanned may not be very different in certain circumstances.

## Planned or unplanned tests

We will focus also on a delineation that many statistical texts emphasize regarding the right MCT—whether the comparisons are planned (i.e., *a priori*) or unplanned (i.e., *post hoc*). Planned comparisons are overwhelmingly recommended for several reasons. First, planned comparisons often result—but not always—in a number of comparisons that is lower than unplanned comparisons. In this case, fewer comparisons may mean (depending on the test) less adjustment to the EER and therefore a less strict threshold of significance. Planned comparisons also ensure that only meaningful and interesting hypotheses are entertained, and that EER is not being adjusted based on uninteresting or meaningless comparisons. Some sources even go so far as to remind us that all statistical designs under a null hypothesis significance testing paradigm should be planned (*Kruschke, 2013*), and different $p$-values and inferences can be developed based on the same data set (through simply changing the sampling intentions and thus critical values used in test statistics).

## Unplanned comparisons

Although planned comparisons are recommended and often provide benefits, the reality is that experimental design and data collection often have surprises, and we cannot always plan everything perfectly. Unplanned comparisons may not include all pairwise combinations; however, they often do include all combinations. Unplanned comparisons for parametric data are by far the most commonly used MCTs, and also the category that offers the greatest variety of MCTs.

As a place to start with unplanned parametric MCTs, some recommend considering Scheffé's $S$ test first (*Ruxton & Beauchamp, 2008*). Although Scheffé's $S$ test is known to be conservative and it is "entirely coherent with ANOVA results," Scheffé's $S$ test is also referred to as a protected test (protected from any differences or inconsistencies with ANOVA results), because a non-significant ANOVA will never produce a pairwise difference in a Scheffé's $S$ test. It is worth mentioning here that it may be surprising to know that the data used in a non-significant ANOVA could still produce a significant pairwise difference in a test other than Scheffé's $S$. *Ruxton & Beauchamp (2008)* go so far as to suggest that "If any procedure other than Scheffe's is used, then it should be implemented regardless of the outcome of the ANOVA." Although there are reasons to consider Scheffé's $S$ test, it should be noted that the test is inherently conservative because it is designed for linear combinations of means and just pairwise comparisons. If no linear combinations are used, Scheffé's $S$ test may be more conservative than desired; however, if linear combinations of means are being compared, it is often an ideal MCT.

If Scheffé's $S$ test is not desired, other options exist. The Bonferroni and sequential Bonferroni tests are commonly used MCTs. Bonferroni is known to be very conservative, and the sequential Bonferroni (*Holm, 1979*) was developed as an alternative approach

which is to be no less conservative. The Dunn-Šidák test (or Šidák correction; *Šidák, 1967*) is a recommended alternative to Scheffé's *S* test. Perhaps most commonly used is Tukey HSD (honest significant difference) test widely applied for parametric unplanned comparisons (*Tukey, 1949*), although the lesser known Tukey-Kramer test (*Kramer, 1956*) should be used in cases of unequal sample size. Our simulations found Tukey's HSD test to be less conservative than the Dunn-Šidák test, and with lower type II error rates than Bonferroni. Overall, the Tukey HSD test is a robust, commonly available, and generally recommended test. Finally, the Waller-Duncan *k*-test (*Waller & Duncan, 1969*) and Ryan's test (*Ryan, 1960*) are less common options, but may be attractive due to their ability to be modified for heterogeneity of variance.

We have excluded recommending Fisher's LSD, Duncan's MRT, or SNK for unplanned comparisons as they are known for not adjusting the EER enough (*Day & Quinn, 1989*; *Ruxton & Beauchamp, 2008*). Specifically, with the SNK test, the EER can become greater than the chosen significance probability when there are two groups of means, such that the means within each group can be equal but the test will say that the groups differ.

## Other considerations
### Interactions
Frequently, research hypotheses may include the consideration of the interactions of fixed effects within a model, be structured as multilevel models, or require interpretation of fixed effects interactions in evaluating hypotheses of categorical data (e.g., extensions of contingency tables, counts or proportions of survey responses, relative abundances of taxa; *Faraway, 2006*; *Agresti, 2015*; *Agresti, 2018*). Although in some cases, these interactions are better modeled by the use of random variables (*Faraway, 2006*; *Zuur et al., 2009*), in other cases, specific research hypotheses, specific models (e.g., multi-category logit), or insufficient sample sizes for parameterization of random effects result in situations where complex interactions of fixed effects are employed.

Interactions may be interpreted in a number of ways, including: directly from parameter estimates, effect sizes estimated from parameter estimates (*Cortina & Nouri, 2000*; *Pituch & Stevens, 2015*), construction of line plots connecting effect means (*Dowdy, Wearden & Chilko, 2003*; *Kutner, Nachtsheim & Neter, 2004*; *Mendenhall III, Beaver & Beaver, 2013*), MCTs, or by a combination of these approaches. Each approach has advantages and disadvantages, and a thorough review is outside of the scope of this effort. However, a few generalizations may help in placing MCTs in the context of these other approaches. Direct interpretation of parameter estimates uses the estimates themselves in a common scale (unit change in *y* given unit change in *x*) for comparison by taking advantage of the sum-to-zero principle in linear models (i.e., differences among levels within and among variables can be compared directly with a variable level set to 0), which is particularly useful for planned comparisons (*Kéry & Royle, 2016*). Determining whether the magnitude of the parameters estimate differences between or among the estimates within an interaction has meaning often uses criteria, such as *p*-values or may use an objective function (i.e., whether AIC suggests inclusion of the interaction term). Performance and concerns over the use of these criteria in this manner are well documented in the literature (*Stephens et al., 2005*;

*Stephens, Buskirk & Del Rio, 2007*; *Murtaugh, 2014*; *Wasserstein & Lazar, 2016*; *Wasserstein, Schrim & Lazar, 2019*). Effect sizes are estimated from the magnitude of differences between or among parameter estimates and offer an alternative criteria, often in units of standard deviation (Cohen's $d$) or variance (Nagelkerke's $R^2$ or $\eta^2$) for assigning meaning to the interpretation of interactions either by the use of published guidelines (e.g., *Cohen, 1988*) or comparison with discipline-specific literature values (*Pituch & Stevens, 2015*). The use of effect sizes addresses some concerns about the use of $p$-values or objective functions (e.g., AIC) to assign meaning (*Lenth, 2001*; *Nakagawa & Cuthill, 2007*; *Ellis, 2010*); however, some effect size estimators may not translate well across applications and are highly influenced by sample size, and importing guidelines across disciplines may be problematic (*Osenberg, Sarnelle & Cooper, 1997*; *Nakagawa & Cuthill, 2007*; *McCabe et al., 2012*; *Pituch & Stevens, 2015*; *Pogrow, 2019*). Graphical interpretation of interactions by plotting lines connecting means is commonly available in software and has a long history of use. Lines may be plotted for any two variables in an interaction, and for higher level interactions, lines may be plotted for two variables controlling for a third or more variables. Similar to interpreting linear models, the slope of the line and whether lines cross among levels of the variable provides meaning (*Dowdy, Wearden & Chilko, 2003*; *Kutner, Nachtsheim & Neter, 2004*; *Mendenhall III, Beaver & Beaver, 2013*) and may be combined with information from $p$-values, effect sizes, or both. The disadvantages are few with the primary problems limited to potentially a large number of plots to examine and that if the magnitude of the difference is of interest, line plots will need to be complemented with parameter estimates criteria (e.g., $p$-values and objective functions) and/or effect sizes. MCTs offer a straightforward method to interpret interactions providing a magnitude of difference in the original (linear models) or link transformed (generalized linear models) units and an adjusted $p$-value. Effect sizes may be readily estimated from the differences, if of interest, and lines may be plotted among the MCT estimated levels. Further, MCTs may be arranged in order of differences with lines or letters indicating significant differences. The disadvantages of MCTs are that software may produce MCTs for uninformative or nuisance comparisons (but see *Steegen et al., 2016*; *Gelman, 2017*), and the choice of MCT heavily influences the outcomes.

Few statistical texts and course resources provide examples of interpreting MCT in the context of complex interactions of multiple fixed effects (but see *Milliken & Johnson, 2001*; *Milliken & Johnson, 2009*; *Gbur et al., 2012*; *Pituch & Stevens, 2015*). Potentially, this could be because authors intended for direct interpretations of parameter estimates, as *Kéry & Royle (2016)*, presented planned comparisons, or interactions to be used in an analysis of covariance (ANCOVA/MANCOVA; *Pituch & Stevens, 2015*), which have little reason for MCT. In our experience, this causes some confusion among students and early career professionals. Therefore, the remainder of this section briefly reviews best practices for applying MCT in complex, multilevel interactions of fixed effects in a model.

Interactions of fixed effects are sometimes seen as a nuisance, rather than as an opportunity for nuanced and detailed understanding of differing levels of categorical fixed effects (hereafter factors) and covariates. Often the simpler, main fixed effects and two-way interactions present attractive and simple interpretations. Yet, three-way and

higher interactions may be fundamental to the study design or may account for important sources of variation. The lack of clarity in the interpretation of a complex interaction in a typical ANOVA-style output table in software (i.e., so which factor made it significant?) and often confusing presentations of individual parameter estimates that may differ in sum-to-zero or sum-to-last based on software choices (i.e., where is the parameter estimate for level $j$?) add to the view of complex interactions as frustrating. MCTs can be very useful in disentangling statistical significance and differences among parameter estimates.

The proper MCT implementation in the case of complex interactions is to perform the MCT on the most complicated statistically significant interaction. In other words, if the model includes three main fixed effects ($X_{i1}$, $X_{i2}$, and $X_{i3}$), three two-way interactions ($X_{i12}$, $X_{i13}$, and $X_{i23}$), and one three-way interaction ($X_{i123}$) that is statistically significant, the MCT should be performed on the three-way interaction ($X_{i123}$). This is because the parameter estimates for $X_{i1}$, $X_{i2}$, $X_{i3}$ $X_{i12}$, $X_{i13}$, and $X_{i23}$ are only meaningful as components of $X_{i123}$ and are not interpretable on their own. If the interaction $X_{i123}$ is not statistically significant, removing the interaction and refitting the model may be warranted (e.g., *Faraway, 2006*) and interpreting the next most complicated interaction. The guiding principle should be *outside-to-inside*, if considering model notation, or *bottom-to-top* if considering an ANOVA-style output table.

Interpreting MCTs in interactions can initially be intimidating, however, understanding the components makes the process easier and indicates where MCT choice impacts the interaction. The output of an MCT will generate an estimate based on the model for each factor ($i$) and level ($j$) of that factor. Although the term *level* is used here, level does not imply that the factor is ordinal or has a numeric value, rather, level is simply used as a convention as the level could be nominal as well. This estimate will be typically accompanied with a $t$-statistic and a $p$-value testing against a null estimate of 0. This $p$-value will be adjusted by the MCT, and if the $p$-value is used to make a determination of statistical significance of the estimate, the choice of MCT is important. If performed, for each pairwise comparison, a difference between estimates, test statistic, and an associated $p$-value are produced. In these comparisons as well, the choice of MCT will affect the test statistic and how the $p$-value is calculated. Sometimes, a comparison will be reported as non-estimable, which may mean that one combination of factor and level is missing or may be insufficiently replicated to generate an estimate. Therefore, when interpreting interactions, one should consider the appropriateness of the MCT for the data and model.

Generally, for two-way interactions, MCT comparisons among the levels of each fixed effect are rather easy to follow. For example, a model with $X_{i1}$, $X_{i2}$, and $X_{i12}$ is presented (Table 4). The sign of the difference in MCT indicates which combination of variable and level is greater and illustrates the directionality of differences. The statistical significance of the MCT adjusted $p$-value indicates where the differences among variables and levels occurred, which is again not evident in ANOVA-style tables. It should be noted that the comparisons are interchangeable in two-way interactions (i.e., although presented as $X_{i1}$, $j = 1$ $-X_{i2}$, $j = 1$, to compare $X_{i2}$, $j = 1$ $-X_{i1}$, $j = 1$, one simply reverses the sign). Thus, a statistically significant two-way interaction of $X_{i12}$ that has a statistically significant, positive $X_1$, $j = 1$ $-X_2$, $j = 1$ comparison would be interpreted as the estimate at level

**Table 4** **Example of a MCT in a two-way interaction.** Variable $X_{i1}$ has 3 levels ($j = 3$) and variable $X_{i1}$ has 2 levels ($j = 2$). Levels are presented as numbers in this example, but also may be words or characters. The method of estimating each variable-level combination (e.g., $X_{i1}, X_{i2}$) depends on MCT, as does the test-statistic.

| Variable $X_{i1}$ Level | Variable $X_{i2}$ Level | Difference |
|---|---|---|
| 1 | 1 | Estimated $X_{i1,j=1}$ − Estimated $X_{i2,j=1}$ |
| 1 | 2 | Estimated $X_{i1,j=1}$ − Estimated $X_{i2,j=2}$ |
| 2 | 1 | Estimated $X_{i1,j=2}$ − Estimated $X_{i2,j=1}$ |
| 2 | 2 | Estimated $X_{i1,j=2}$ − Estimated $X_{i2,j=2}$ |
| 3 | 1 | Estimated $X_{i1,j=3}$ − Estimated $X_{i2,j=1}$ |
| 3 | 2 | Estimated $X_{i1,j=3}$ − Estimated $X_{i2,j=2}$ |

1 of $X_{i1}$ is significantly greater than level 1 of $X_{i2}$. One also can compare across levels; a statistically significant $X_{i1}, j = 1 - X_{i2}, j = 2$ would be interpreted as level 1 of $X_{i1}$ differs from level 2 of $X_{i2}$. These comparisons do require an appreciation of the conditional nature of the comparison, for example, $X_{i1}, j = 1 - X_{i2}, j = 2$ does not mean that levels 1 and 2 are different always, they mean that levels 1 and 2 differ when comparing $X_{i1}$ with $X_{i2}$ and may not apply if the model includes another factor, $X_{i3}$.

In higher level interactions, such as three-way or higher (e.g., $X_{i123}$), the pairwise comparisons (e.g., $X_{i2}, j = 1 - X_{i1}, j = 1$) are conditional on other variables (e.g., $X_{i3}$) in the interaction. Continuing with the example model with $X_{i1}$, $X_{i2}$, $X_{i3}$, $X_{i12}$, $X_{i13}$, $X_{i23}$, and $X_{i123}$, a three-way interaction is presented (Table 5). Using the first comparison in the table, a statistically significant comparison of $X_{i2}, j = 1 \mid X_{i1}, j = 1 - X_{i3}, j = 1 \mid X_{i1}, j = 1$ would be interpreted as conditional on the first level of $X_{i1}$, level 1 of $X_{i2}$ differs from level 1 of $X_{i3}$. If the variable $X_{i1}$ represents a sampling location, this pairwise comparison would be interpreted to indicate that $X_{i2}$ level 1 differs from $X_{i3}$ level 1 at the first sampling location. Higher level interactions of more than three variables follow the same logic, where each additional variable adds another conditional influence.

Not all comparisons in MCT will be logical or relevant to the hypothesis being investigated. For example, the hypothesis could be investigating different food items in predator diets between seasons; however, the data were collected in different rivers, thus, resulting in a two-way interaction, *river* x *season*. In this case, *river* is included because there are inherent differences that cause variation. Ignoring this variation would be a mistake resulting in improper estimation of parameters and error, therefore, *river* is included in the interaction. However, only MCT results regarding *season* are relevant to the hypothesis. Although following *Steegen et al. (2016)* and *Gelman (2017)*, there is value in examining in the "multiverse" of multiple comparisons, it is still up to the investigator to focus on the relevant comparisons.

## GLMs

Generalized linear models (GLMs) present a highly useful class of models applicable in a number of situations, given that a link function describes the linear relationship between the observed mean with the mean of the linear combination of the model and the response
**Table 5  Example of a MCT in a three-way interaction.** Variable $X_{i1}$ has 2 levels ($j = 2$), variable $X_{i2}$ has 2 levels ($j = 2$), and variable $X_{i3}$ has 2 levels ($j = 2$). The notation **variable 1|variable 2** indicates the estimate is conditional on the second variable. For all situations, the test statistic and adjusted $p$-value depends on the choice of MCT.

| Variable $X_{i1}$ Level | Variable $X_{i2}$ Level | Variable $X_{i3}$ Level | Difference |
|---|---|---|---|
| 1 | 1 | 1 | Estimated $X_{i2,j=1}|X_{i1,j=1}$ − Estimated $X_{i3,j=1}|X_{i1,j=1}$ |
| 1 | 1 | 2 | Estimated $X_{i2,j=1}|X_{i1,j=1}$ − Estimated $X_{i3,j=2}|X_{i1,j=1}$ |
| 1 | 2 | 1 | Estimated $X_{i2,j=2}|X_{i1,j=1}$ − Estimated $X_{i3,j=1}|X_{i1,j=1}$ |
| 1 | 2 | 2 | Estimated $X_{i2,j=2}|X_{i1,j=1}$ − Estimated $X_{i3,j=2}|X_{i1,j=1}$ |
| 2 | 1 | 1 | Estimated $X_{i2,j=1}|X_{i1,j=2}$ − Estimated $X_{i3,j=1}|X_{i1,j=2}$ |
| 2 | 1 | 2 | Estimated $X_{i2,j=1}|X_{i1,j=2}$ − Estimated $X_{i3,j=2}|X_{i1,j=2}$ |
| 2 | 2 | 1 | Estimated $X_{i2,j=2}|X_{i1,j=2}$ − Estimated $X_{i3,j=1}|X_{i1,j=2}$ |
| 2 | 2 | 2 | Estimated $X_{i2,j=2}|X_{i1,j=2}$ − Estimated $X_{i3,j=2}|X_{i1,j=2}$ |
| 1 | 1 | 1 | Estimated $X_{i1,j=1}|X_{i2,j=1}$ − Estimated $X_{i3,j=1}|X_{i2,j=1}$ |
| 1 | 1 | 2 | Estimated $X_{i1,j=1}|X_{i2,j=1}$ − Estimated $X_{i3,j=2}|X_{i2,j=1}$ |
| 2 | 1 | 1 | Estimated $X_{i1,j=2}|X_{i2,j=1}$ − Estimated $X_{i3,j=1}|X_{i2,j=1}$ |
| 2 | 1 | 2 | Estimated $X_{i1,j=2}|X_{i2,j=1}$ − Estimated $X_{i3,j=2}|X_{i2,j=1}$ |
| 1 | 2 | 1 | Estimated $X_{i1,j=1}|X_{i2,j=2}$ − Estimated $X_{i3,j=1}|X_{i2,j=2}$ |
| 1 | 2 | 2 | Estimated $X_{i1,j=1}|X_{i2,j=2}$ − Estimated $X_{i3,j=2}|X_{i2,j=2}$ |
| 2 | 2 | 1 | Estimated $X_{i1,j=2}|X_{i2,j=2}$ − Estimated $X_{i3,j=1}|X_{i2,j=2}$ |
| 2 | 2 | 2 | Estimated $X_{i1,j=2}|X_{i2,j=2}$ − Estimated $X_{i3,j=2}|X_{i2,j=2}$ |
| 1 | 1 | 1 | Estimated $X_{i1,j=1}|X_{i3,j=1}$ − Estimated $X_{i2,j=1}|X_{i3,j=1}$ |
| 1 | 2 | 1 | Estimated $X_{i1,j=1}|X_{i3,j=1}$ − Estimated $X_{i2,j=2}|X_{i3,j=1}$ |
| 2 | 1 | 1 | Estimated $X_{i1,j=2}|X_{i3,j=1}$ − Estimated $X_{i2,j=1}|X_{i3,j=1}$ |
| 2 | 2 | 1 | Estimated $X_{i1,j=2}|X_{i3,j=1}$ − Estimated $X_{i2,j=2}|X_{i3,j=1}$ |
| 1 | 1 | 2 | Estimated $X_{i1,j=1}|X_{i3,j=2}$ − Estimated $X_{i2,j=1}|X_{i3,j=2}$ |
| 1 | 2 | 2 | Estimated $X_{i1,j=1}|X_{i3,j=2}$ − Estimated $X_{i2,j=2}|X_{i3,j=2}$ |
| 2 | 1 | 2 | Estimated $X_{i1,j=2}|X_{i3,j=2}$ − Estimated $X_{i2,j=1}|X_{i3,j=2}$ |
| 2 | 2 | 2 | Estimated $X_{i1,j=2}|X_{i3,j=2}$ − Estimated $X_{i2,j=2}|X_{i3,j=2}$ |

is a random variable belonging to the exponential family of distributions. As a useful class of models, MCT techniques have been applied with GLMs. MCT is commonly and correctly used on the GLM estimates in their link transformed values; however, some confusion may occur when the MCT results are compared with the values in the original space (i.e., estimates inverse link transformed), particularly in cases where the interpretation based on visual inspection would differ (e.g., MCT on the GLM estimates suggests a difference not obvious in the observed data and vice versa). By the principle of invariant reparameterization of maximum likelihood estimates, statistical significance of MCT in GLM confers statistical significance in the observed data as well.

## False discovery rates

In addition to the previously mentioned alternatives to MCT (direct interpretation of parameter estimates, effect size estimation, and line plots), False Discover Rates (FDR) are another widely used method for interpreting meaningful statistical outcomes through an alternative process to control type I error rates. Following *Benjamini & Hochberg (1995)*

and *Verhoeven, Simonsen & McIntyre (2005)*, MCT can be thought of as controlling the chance of at least one type I error at a desired level across all tests (V), whereas FDR can be thought of as controlling the proportion of type I errors across all significant tests (i.e., V/r or desired rate/discoveries). FDR has the advantage of greater statistical power than traditional MCTs (i.e., keeps type II level higher than a MCT across the same number of tests; *Garcia, 2004*; *Garcia, 2005*), being adaptive (i.e., rate is based on the number of discoveries, rather than number of overall tests; *Garcia, 2005*; *Verhoeven, Simonsen & McIntyre, 2005*), and being consistent (also called scalable, i.e., rate has the same meaning, regardless of the number of discoveries). Recent comparisons of MCT (family-wise error rate methods) with FDR methods demonstrate the advantages of FDR; however, also demonstrated the importance of proper implementation (*Brinster et al., 2018*; *White, Van der Ende & Nichols, 2019*). *Verhoeven, Simonsen & McIntyre (2005)* suggest that FDR is relatively simple to implement, even in a spreadsheet, although computational mistakes occur in the literature (*White, Van der Ende & Nichols, 2019*). Further, FDR introduces new considerations in reporting (e.g., adjusted *p*-value vs. *q*-value, with or without effect sizes and confidence intervals), and rates from FDR are not directly comparable to previously reported literature complicating comparisons. Therefore, although FDR offers numerous advantages, the goals of particular study may not be compatible for FDR alone suggesting a role for MCTs in studies.

## CONCLUSIONS

### Recommendations

The field of multiple comparisons includes a wide variety of very necessary procedures that often directly contribute to the results of scientific studies. Despite myriad test options, certain tests remain more popular than others, while some tests are rarely used. Due to the complexity of MCT choices and the increasing diversity of data and models that are being used, it is not realistic to come up with a one-size-fits-all approach for their application. In many cases, identifying the basics of planned vs. unplanned comparisons and parametric vs. non-parametric data and models will narrow down MCT options (Fig. 7). In addition to a decision tree approach for selecting MCTs, we have identified some broad recommendations from our own work and borrowing from others.

1. Know that you often have choices when it comes to MCTs. Although some data situations will leave you with only one test, many data and models will have more than one MCT to choose from.

2. Do not include more comparisons than you need. Consider each comparison to be a hypothesis. When extra and uninteresting comparisons are included, they not only provide no scientific progress, but they also add to the error rate adjustment by increasing the threshold for significance for other comparisons.

3. As noted by others, avoid Fisher's LSD, Duncan's MRT, and the SNK tests. These tests are very liberal as they do not make acceptable error rate adjustments.

4. For parametric situations, Scheffé's *S* test is coherent with ANOVA and especially recommended for linear combinations of means (not just pairwise comparisons).

However, absent linear combinations of means, Tukey's HSD presents a robust and widely available test for a variety of situations.

5. When selecting an MCT, even the recommended MCTs perform differentially among studies with large and small observations with many and few groups. It may be necessary to compare among MCTs to determine the MCT that best suits the number of groups and number of observations within each group in a particular study.

6. Consider that MCTs may not be the only option in a particular study. Other approaches, such as direct interpretation of the parameter estimates, effect size estimation, and plots covered in the 'Interaction' section or FDR, discussed previously, may be complementary or alternative options. Also consider that MCTs are closely associated with $p$-values, and numerous critical evaluations of $p$-values suggest $p$-values alone (or at all) may be insufficient for interpretation.

## ACKNOWLEDGEMENTS

This manuscript was approved for publication by the Director of the Louisiana Agricultural Experiment Station as MS 2020-241-34918. This is contribution no. 2020-24 of the Quantitative Fisheries Center, Michigan State University. We also thank Patrick Midway for design assistance on the decision diagram.

### Funding

This material is based upon work that also was partially supported by the National Institute of Food and Agriculture, U.S. Department of Agriculture, under the McIntire-Stennis Cooperative Forestry Program as project number LAB-94171. There was no additional external funding received for this study. The funders had no role in study design, data collection and analysis, decision to publish, or preparation of the manuscript.

### Grant Disclosures

The following grant information was disclosed by the authors:
National Institute of Food and Agriculture.
McIntire-Stennis Cooperative Forestry Program: LAB-94335.

### Competing Interests

The authors declare there are no competing interests.

### Author Contributions

- Stephen Midway and Matthew Robertson conceived and designed the experiments, performed the experiments, analyzed the data, prepared figures and/or tables, authored or reviewed drafts of the paper, and approved the final draft.
- Shane Flinn and Michael Kaller analyzed the data, prepared figures and/or tables, authored or reviewed drafts of the paper, and approved the final draft.
## Data Availability

All data and code are available at GitHub: https://github.com/stevemidway/MultipleComparisons.

## Supplemental Information

Supplemental information for this article can be found online at http://dx.doi.org/10.7717/peerj.10387#supplemental-information.

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
