# Peer review of "Comparing multiple comparisons: practical guidance for choosing the best multiple comparisons test"

_PeerJ, doi:10.7717/peerj.10387_

## Round 0.1 · original submission · Major Revisions

Please make sure to pay particular attention to the Reviewers comments about discussing other approaches for making conclusions about meaningful effects, including focusing on parameter estimation rather than p-values and false discovery rate approaches. With respect to the latter, consider similar discussions of the use of FDR procedures in Biology (e.g.: Verhoeven et al. 2005 https://doi.org/10.1111/j.0030-1299.2005.13727.x)

Reviewer 1 ·

Basic reporting

The authors have reviewed the literature to show which types of multiple comparisons tests (MCTs) are being used by authors of papers in ecological journals (and just as important, which are not being used), and then present results of a simulation study to compare the performance of different MCTs. The literature review provides nice context and seems to be done in a reasonable way, but by itself would not provide sufficient content for an interesting publication. The simulation results are the more substantial part of the manuscript. I have looked at the authors’ code and results, and see no flaws. Their main results repeat the findings from some earlier papers; a few methods for MCTs are fundamentally flawed and should not be used, whereas a few others perform well. Nonetheless, I am not aware of any previous papers that are as thorough or clear. Thus, I think this work would provide a useful reference for many researchers, including those beyond the ecology researchers that are the focus of the literature review.

The manuscript is well written. I found the text clear and free of grammatical errors. The authors did a good job proof-reading and preparing the manuscript, which made the task of reviewing simpler. Thanks.

My main recommendation for improving the manuscript would be to expand the discussion of methods to at least touch upon the methods for false discovery rate (FDR) adjustments. The Benjamini-Hochberg method for family wise error rate adjustments (FWER, line 116) has become extremely common in other sub-disciplines of biology over the last couple decades, and is used at least as commonly as MCTs. I teach graduate biostatistics and draw students from a range of fields. Because of its importance in “-omics” analyses, I have had to add discussion of FDR approaches alongside older MCTs such as Bonferroni, Tukey HSD, Scheffé, and Dunn-Šidák (the only MCTs I covered the last time I taught biostatistics). Despite being dominant in a slightly different field, I think that most biology researchers and students are likely to be familiar with (or at least have seen reference to) the Benjamini-Hochberg FDR approach. Thus it would greatly improve the relevance and utility of the present manuscript to include some discussion of the relationship between FDR and MCT methods.

Experimental design

The simulations are well documented as I saw no errors in the code.

Validity of the findings

No comment.

Reviewer 2 ·

Basic reporting

This is a useful study, but is narrowly focused, so that results are limited in scope. The English is mostly very good, but there are a few poorly phrased sentences where the meaning would not be clear to the intended reader, and there are also some erroneous statements that need to be corrected. A wider range of relevant literature could have been used.

Experimental design

The design of the simulations is adequate, but limited in scope, and as a result the results are not as general as they could be.

Validity of the findings

The results are sometimes not interpreted well, with some minor errors. Much of the paper is advice drawn from previous literature, rather than based on results presented here. I consider that the advice should be more balanced, covering the various interpretations of best practice that are present in the literature.

Additional comments

The paper first shows that despite good advice in published papers and data analysis texts written for ecologists, there are still many ecological papers published in which the multiple comparison tests used are poorly chosen. It then provides advice to ecologists as to which tests should be used. This is actually the most difficult part. Perhaps the visual presentation in the paper of the way these tests behave will grab attention to the issue and further research by future authors – I hope so. But to quote from Oaten (1995) Ecology 76: 2001-9: “The difficulties arise from the need to present the methods concisely and, as far as possible, painlessly. …. assertions tend to rely on proof by authority and to take the form of exhortations and instructions rather than statements of results…In time, some judgments become accepted as laws. These can be hard to challenge”. Further, simulations can be misleading, by showing what happens only under particular conditions and with particular types of data.
The best starting point is probably to note that most statisticians would encourage estimating confidence limits for the size of effects (e.g. differences between means), rather than use significance tests, because using tests encourages readers to mistakenly believe that effects are real if (and only if) the test is significant. MCTs can be seen as an attempt to maintain this belief even when researchers are actually data-snooping rather than testing pre-set hypotheses. Thus, I would have preferred to see this paper start with a brief summary to point out that ideally ecologists should be setting out to estimate the size of hypothesised effects, rather than testing whether their data provides a significant result for an effect. They should be especially cautious if the study design involves questions about which of several groups may differ, and try if possible to ask pre-set questions about particular differences, on the basis of the focus of the study or the theory that drives the study. Often these planned questions will not relate to differences between pairs of means, but more complex differences. The best approach is to estimate the size of these differences (with confidence limits). MCTs appear to be appropriate when either the investigator has little idea of which groups may differ when setting up the study, or the results of a study are surprising, so that new, unplanned questions arise after examining the data. At this point, one could enumerate the comparisons that have reasonable biological interpretations and set up MCTs based on this set of comparisons. However, an alternative view is that the PCER could be used for all comparisons, with a warning to the reader that the overall error rate would be >0.05 as each question is considered in isolation from the others.
My other general concerns with this paper are that:
1. The data used for simulations is all drawn from the normal distribution, with equal variances. This is a missed opportunity, as it is very unlikely that most data from ecological experiments (or in many other types of experiments or surveys) would be normally distributed (in many cases, not even very well behaved), or have equal variances in each treatment, as authors often do not consider the likely distribution of their data. A study that included the robustness of the MCTs to modest differences between variances and modest variations in data distribution (modest meaning those which would not be obvious in small samples) would have been more useful, as it would show the behaviour of tests in the context of the kind of data that they are used on in practice, and as previous studies have examined the error rates of most of these tests on normally distributed data with equal variances. Further, the issue of unequal variances is not covered in the recommendations, especially in regard to the use of non-parametric tests (see my specific comment below).
2. The authors essentially dodge the issue of what the criteria are for using non-parametric tests. Many researchers feel that if they are not sure if the data are normally distributed, they should use a non-parametric test, and this is not good practice, partly because the issue of unequal variances requires a different analysis process, and partly because parametric tests (which actually rely on the distribution of the sample mean being approximately normal) are valid for moderately non-normal samples even for small sample sizes. Tests on samples that have different skewed distributions are a case where rank-based tests such as the MWW-U test are useful because the use of ranks reduces the effects of extreme observations, but there are also other ways to reduce the effect of extreme observations when using parametric tests. The advantage of this is that estimates of differences are possible.
3. I initially found it hard to understand why the simulated type 1 error rate of the parametric tests in Figure 3 is far below the 5% level in all cases. Then I realized that the error rates shown are per comparison rates, although this is not specified in the text or in the figure captions. If you want to know how well a test procedure works, it is more useful to show how well it does what it is designed to do, which is to ensure the EER rate does not exceed the 5% rate. So I would want to see the Type 1 EER for each test. I suggest this ought to be shown, and in any case the caption should make clear which error rates are shown. Obviously, the tests control EER by reducing the PCER rate below 5%. The results report that the study designs with many groups had the lowest per comparison error rates, with no explanation. This is what the MCTs must do, to control the EER.
4. The investigation of type 1 error only when a group of means are all equal misses the problem that occurs with the SNK test, where the EER becomes greater than the chosen significance probability when there are two groups of means, such that the means within each group are equal, but the groups differ (see Quinn, G.P. and Keough, M.J. (2002) Experimental design and data analysis for biologists. Cambridge University Press, Cambridge, p200). This leads to a false view of the performance of this test in Figures 3 and 4. As this was a known problem with this test, I suggest the authors should have demonstrated this problem.
5. It seems strange that the paper records the type 2 error rates of the various parametric MCTs (when one group mean differs, using normal distributions, for 4 study designs), but these results are not used when discussing which tests are recommended for various situations. The results (in Figure 5) show for example, that there is consistently less Type 2 error for Tukey’s HSD test than for the Bonferroni and Dunn-Sidak tests that are used for equivalent situations. While the difference is small, this suggests that the HSD tests will often be more powerful and thus a better choice. Exceptions might occur if the data are not of the type simulated, but this result conforms to that shown in Day and Quinn 1989, where the pattern of differences between groups was different, which suggests this is likely to be a consistent advantage of the HSD. On the other hand, the HSD can only be used with pairwise comparisons, while the Bonferroni and Dunn-Sidak methods were designed for any (potential) number of comparisons, whether pairwise or not.
6. The penultimate part of the discussion discusses using MCTs to interpret interactions in complex ANOVA designs. “MCTs can be very useful in disentangling statistical significance and differences among parameter estimates” (Lines 354-5). Many statistical texts would disagree (for example, Quinn and Keough 2002, p252), and I think the alternate approaches should be mentioned, rather than imply this is always the best or only way to untangle interaction effects. The problem here is that there will often be a large number of means (so that the MCTs are not powerful), and the results of the many MCTs can be ambiguous. A simple example is where there is a significant interaction shown by the ANOVA, yet no pairwise comparison is significant. This manuscript itself points out that the MCTs may not be consistent with the ANOVA (lines 311-3). I suggest it makes more sense to try to understand why the interactions occur using interaction plots of means (a line joining the means of level 1 of factor A at each level of factor B, another line joining the means of level 2 of A at each level of B, etc). Three way or more complex designs can be broken down - for example plots of each two-way interaction at each level of a third factor. Another method is to construct sums of squares to test the effect of factor A separately at each level of Factor B (or the effects of factors A and B at each level of C where there is a three-way interaction, etc). These tests would use the mean-square error of the original ANOVA, which is the best estimate of the error variance, with more degrees of freedom (and thus more power) than if the data were separated to run these analyses. Tests of fixed factors would normally be run at each level of a random factor (for example ‘river’ in the example in lines 403-7), but with two fixed factors one might test the effect of A at each level of B, and the effect of B at each level of A. Plots of the means for fixed factors at each level of the other factors would aid in interpretation. As these are exploratory analyses looking for significant effects in a series of tests (especially if there are many levels of a factor), a Bonferroni or Dunn-Sidak adjustment of the significance levels may be appropriate, depending on how the results are reported to the reader. But the number of tests involved would almost certainly be less than MCTs on all the means involved in an interaction.
7. The Abstract does not include some of the most useful conclusions presented in the discussion. I suggest these may include that Planned comparisons are overwhelmingly recommended (lines 262-3) and an unadjusted significance level can be used, that for planned non-parametric comparisons the Mann-Whitney -Wilcoxon U test is recommended, and that if planned comparisons are not used, Scheffe’s test may be used for any linear combination of the means, and Tukey’s HSD, the Bonferroni or the Dunn-Sidak tests are most commonly used for pairwise comparisons of groups. Other tests are recommended for particular types of data.

Specific comments.
Abstract:
Lines 19-20: The meaning of: “Due to the variable conditions of the data being analysed” is not clear, and this clause should be rephrased to make the meaning clear to readers.
Line 22: “including >40,000 reports of their use in ecological journals” does not convey any information – there is no time period specified. Perhaps a statement such as: “and we have documented extensive use of MCTs in ecological research” would be better.
Line 22: MCT should be MCTs.
Line 24: “We first reviewed the recommendations on their correct use.” There is nothing about this in the Results, Methods or the Background section of the Introduction. I assume that many of the recommendations in the discussion are taken from this review. I would have expected to see how this review was done – were a range of statistics texts consulted? Or were the initial descriptions of the tests checked to see what they were designed to do? Or what? And some MCTs were designed or recommended for particular situations. The only reference to this is the statement “Principles behind matching a MCT to an experimental design are discussed below.” on line 174 in the Methods. ‘Below’ turns out to be in the Discussion. But this should be in the results before the results of simulations, to help the reader evaluate what the simulation results indicate in terms of what to use. One example is Scheffe’s test – see my comment below. Another is the Tukey-Kramer test, used to replace Tukey’s test when the sample sizes are unequal – see my comment below.
Introduction:
Line 34: The meaning of “Data analyses are crowded with factors of interest from experiments and observations” is not clear.
Line 51: “multiple comparisons evaluations” should be “the evaluation of multiple comparisons”.
Line 100: “creates more work for α” is poorly phrased. The non-statistician readers this paper is directed to will not understand what is meant.
Lines 112-113: “The EER reflects the adjustment in error rates to account for multiple comparisons” should be “The EER reflects the adjustment in PCER to account for multiple comparisons” and “ ways to adjust for EER” should be “ways to adjust the PCER” (the EER is a chosen value, so “to adjust for EER” has no obvious meaning.)
Lines 114-5: The statement “EERs that reduce the power to detect differences are known as conservative, while those adjustments that are less strong are known as liberal” simply does not make sense. MCTs that control the EER to below 5% (by strongly reducing PCERs) are known as conservative, while those with less strong adjustments of the PCERs which do not control the EER to at or below 5% are known as liberal.
Methods:
Lines 121-132: The search method is useful, although it has the limitations described by the authors, and the Table of search results interesting. However, there are other issues here. The frequencies with which these methods are used does not tell us whether they were used for the correct types of data. For example, it would be useful to know how often the 3 common methods used to compare adjusted means after an analysis of covariance, as this is a likely error.
Lines 190-192: See my general comments on using normal distributions with equal variances, and on the misrepresentation of type 1 error rates for the SNK in practice, where means are not all equal.
Lines 192-193: Setting only 1 mean to be different from the rest to guage type 2 error may not provide a good indication of the differences in power between some tests, as in practice several means may differ in an experiment.
Line 207: “We excluded the Tukey-Kramer test and Dunnett’s test since they are only applicable for special cases”. But the Tukey-Kramer test is used in place of the Tukey test when sample sizes are unequal (‘unbalanced group study designs’). Does this mean that even for the unbalanced designs the authors evaluated Type 1 and 2 errors for the Tukey test?
Results:
Lines 225-6: Treatment with higher sample sizes did not always have lower per comparison error rates than equivalent designs with lower sample sizes. This is not true of the LSMG versus HSMG for balanced designs, nor of the HSFG versus LSFG in unbalanced designs (for most tests). Nor is it true that unbalanced designs reduced the proportion of per comparison type 1 error for all tests – the reverse is true for many tests in the HSFG design and in the LSMG design.
Lines 228-231: As I would expect, the SNK does not exceed an error rate of 0.05 in Figure 3. But Fig 3 shows it does not always have a higher error rate than the other tests as indicated here. Further, in the p-value density plots of Figure 4, it is Scheffe’s test that is relatively constant from zero to one (for some sample conditions), not the SNK test, which has a clear peak near 1. This error should be corrected.
Figure 4 and Figure 6: The four parts of each of these figures are labelled a, b, c, d and thus the caption statement in both figs that “Simulation group abbreviations can be found in the Figure 2 caption” is of no use to the reader. The simulation group abbreviations should be used in place of the a-d.
Line 231: Scheffe’s S test is designed to test any linear comparison of the means, not just pairwise comparisons, and this is why the test is conservative (reduces the EER below 5%) (and less powerful – see Fig 5) relative to tests designed for use when only pairwise comparisons are considered. It is also why it coherent with ANOVA results, as ANOVA F tests depend on whether any linear combination of the group means is larger than expected by chance. The reader should be made aware of this in the ‘Background’ section, or when the selection of tests was described (line 203), but also reminded of this when discussing its selection for unplanned comparisons (lines 307-315). It is not clear if the use of this test in the surveyed literature was for its designed purpose, or if it has been frequently used for pairwise comparisons. I assume the frequencies shown in Table 2 represent the sum of either use of the test.
Discussion:
Lines 254-259 (Parametric or non-parametric data). This section should also deal with the issue of unequal variances among the groups tested, This is a much more frequent problem in practice in terms of deciding whether a parametric general linear model analysis should be used, and non-parametric methods are also subject to an assumption that the distributions in the treatments compared are identical in shape and scale (i.e. except for the median or mean)- see Johnson, D.H. 1995 Ecology 76: 1998-2000. Thus the variances must be similar (on some scale) for the test to function correctly. In other words, the use of non-parametric tests may be useful when the data is far from normally distributed, but they do not solve the problem of unequal variances – unless some transform of the data would equalise the variances but make the distributions very non-normal. There are methods available that cope with unequal variances – there is mention of this later in the discussion, but no advice on when and why they should be used.
Lines 272-3: Note that orthogonal contrasts are not completely independent statistically (see the Oaten, 1995 reference above).
Lines 273-4: Orthogonal contrasts are not necessarily comparisons that do not include the same group in more than one comparison. In the example of A, B, C and D used, A to B, C to D and (A+B)/2 to (C+D)/2 are all orthogonal contrasts. I cannot see the point of such a “loose definition made operational for non-statisticians” (lines 276-7) nor do I see the value of a conflation of orthogonal comparisons and planned comparisons, even if ‘other literature’ has adopted this (I note that no reference is provided here).
Lines 282-3: “there is little practical difference between planned non-orthogonal comparisons and unplanned non-orthogonal comparisons”. I do see a difference between these, and the key issue is whether the comparisons are planned or not (see Ruxton and Beauchamp 2008) . First, if the limited set of questions that are of interest to the ecologist in setting up the study define comparisons that are not all orthogonal, I see no reason why they should not all be tested using the PCER, provided the logical links between non-orthogonal questions are pointed out to the reader. I see the examination of whether comparisons are orthogonal mainly as a useful check for the researcher, to point out to her/him that the answer to one question may be linked to the answer to others, and as a brake on the number of comparisons made, as the number of potential orthogonal contrasts is the number of groups minus 1. Unplanned comparisons, on the other hand, by definition, may arise from any of the potential ways to compare a set of groups with each other. Only considering pairwise comparisons is one way to reduce the potential number of these unplanned comparisons, yet gain information about where any differences arise among the groups.
Lines 309-311: See my comments on Scheffe’s test above (Results, line 231). It would be useful here to explain when Scheffe’s test is designed to be useful – when a complex comparison is chosen for testing after the results have been examined.
Lines 318-319: Simultaneous testing using the Bonferroni adjustment of the PCER is known to be conservative – that is, it reduces the EER below 5%, and thus it often reduces the PCER below the level that other tests do (see Figure 3). The sequential Bonferroni should be less conservative than the simultaneous test (and the same is true for the sequential Dunn-Sidak test). As shown in Figure 3 the Dunn-Sidak adjustment of PCER is slightly less conservative in some designs, and Tukey’s HSD is less conservative in some cases than the Dunn-Sidak tests. The power of tests is reduced if the tests are more conservative than they need to be, but this is not well shown in Figure 5, because the pattern of differences between the means that was simulated does not show these differences well. One can see however, that the HSD often has slightly lower type 2 error rates than the Bonferroni and Dunn-Sidak tests.
Table 4: The difference in the 3rd row of the table should have j=2 for the first variable.
Table 5: The 9th row repeats the 8th row and should be deleted. There appears to be a 2-1-2 row missing below row 12.
Conclusions:
Surely Figure 7 should be mentioned in these conclusions?
I disagree with the thrust of the recommendation regarding Scheffe’s test, which is designed for an unusual situation of a complex comparison, selected from all such potential comparisons after the results re available, and thus constrains the PCER to very low levels and has very low power compared to other MCTs. Several studies show Tukeys HSD is the pairwise test with the best power, so this is the best recommendation for all pairwise contrasts, unless sequential testing is to be done. While the authors may not agree with my view, I think they should explain the ways the MCTs were designed for different types of comparisons.

---

## Round 0.2 · accepted · Accept

Thank you for your thorough treatment of the reviewers' comments and careful revision.